# Sclerostin as Regulatory Molecule in Vascular Media Calcification and the Bone–Vascular Axis

**DOI:** 10.3390/toxins11070428

**Published:** 2019-07-21

**Authors:** Annelies De Maré, Stuart Maudsley, Abdelkrim Azmi, Jhana O. Hendrickx, Britt Opdebeeck, Ellen Neven, Patrick C D’Haese, Anja Verhulst

**Affiliations:** 1Laboratory of Pathophysiology, Department of Biomedical Sciences, University of Antwerp, 2610 Wilrijk, Belgium; 2Receptor Biology Lab, Department of Biomedical Sciences, University of Antwerp, 2610 Wilrijk, Belgium; 3Translational Neurobiology Group, VIB Center for Molecular Neurology, University of Antwerp, 2610 Wilrijk, Belgium

**Keywords:** chronic kidney disease, vascular calcification, bone disease, mineral abnormalities, rat model

## Abstract

Sclerostin is a well-known inhibitor of bone formation that acts on Wnt/β-catenin signaling. This manuscript considers the possible role of sclerostin in vascular calcification, a process that shares many similarities with physiological bone formation. Rats were exposed to a warfarin-containing diet to induce vascular calcification. Vascular smooth muscle cell transdifferentiation, vascular calcification grade, and bone histomorphometry were examined. The presence and/or production of sclerostin was investigated in serum, aorta, and bone. Calcified human aortas were investigated to substantiate clinical relevance. Warfarin-exposed rats developed vascular calcifications in a time-dependent manner which went along with a progressive increase in serum sclerostin levels. Both osteogenic and adipogenic pathways were upregulated in calcifying vascular smooth muscle cells, as well as sclerostin mRNA and protein levels. Evidence for the local vascular action of sclerostin was found both in human and rat calcified aortas. Warfarin exposure led to a mildly decreased bone and mineralized areas. Osseous sclerostin production and bone turnover did not change significantly. This study showed local production of sclerostin in calcified vessels, which may indicate a negative feedback mechanism to prevent further calcification. Furthermore, increased levels of serum sclerostin, probably originating from excessive local production in calcified vessels, may contribute to the linkage between vascular pathology and impaired bone mineralization.

## 1. Introduction

Cardiovascular disease is responsible for a substantial part of mortality among the elderly and patients with diabetes, chronic kidney disease (CKD), hypertension, and osteoporosis. Vascular media calcification (i.e., so-called arteriosclerosis or Mönckeberg’s sclerosis) is a prominent aspect of cardiovascular disease in these patient populations. Vascular calcification is an active cell-regulated process in which the vascular smooth muscle cells (VSMCs) play a prominent role [1]. 

One of the key events during the vascular calcification process is the transdifferentiation of VSMCs, a process that now potentially affords the capacity for being exploited as an important therapeutic target [2]. Since transdifferentiated VSMCs actively deposit hydroxyapatite in the medial layer, it is generally accepted that they transdifferentiate towards an osteoblast/chondrocyte phenotype. However, since VSMCs originate from mesenchymal stem cells, dedifferentiated VSMCs could theoretically also transdifferentiate towards adipocytes instead of osteoblasts/chondrocytes. The phenotypic fate of cells depends on their exposure to extracellular stimuli, which will activate or inhibit different sets of signaling pathways. Osteoblast and chondrocyte differentiation is under the control of Wnt/β-catenin signaling. Peroxisome proliferator-activated receptor γ (PPARγ) signaling, on the other hand, is a strong inducer of adipocyte differentiation. In this context, it is worth mentioning that a regulating effect of PPARγ signaling on the vascular calcification process has been reported [3,4]. A regulatory protein that acts on both pathways is sclerostin [5,6]. Sclerostin is best known as an inhibitor of canonical Wnt/β-catenin signaling within the bone formation process. However, recent studies showed that sclerostin can also enhance adipocyte differentiation [6]. In this context, it is important to recognize that in recent years, sclerostin has received growing interest with regard to its potential role in the development of vascular calcification, in addition to its effects on bone metabolism. An association between increased serum sclerostin levels and decreased mortality has been observed in hemodialysis patients [7,8]. This is in accordance with the concept that sclerostin may prevent/inhibit vascular calcifications in a manner reminiscent of its inhibitory action on bone formation, thereby reducing mortality. On the other hand, sclerostin, the expression of which until recently was thought to be confined to osteocytes, has now also been demonstrated in VSMCs and valve tissue adjacent to areas of calcification [9,10,11]. Furthermore, CKD patients with aortic calcifications were found to have higher serum sclerostin levels compared with CKD patients without vascular calcification [9,10,12]. In this context, it is worth mentioning that renal elimination of sclerostin increases with declining renal function [13]. Moreover, sclerostin is a positively charged molecule which is easily filtered by the negatively charged glomerular membrane [14]. Taken together, this allows us to put forward the hypothesis that sclerostin, originating from excessive local production in calcified vessels, may spill over to the serum by which it may prevent further progression of (vascular) calcifications and contribute to the high incidence of low bone turnover in CKD patients [15]. 

In general, the precise role of sclerostin in VSMC transdifferentiation and the development of vascular calcifications, as well as its potential role in the bone–vascular axis, is poorly understood. This issue should be of particular clinical interest given the recent therapeutic introduction of an antisclerostin antibody treatment to stimulate bone formation [16]. Since clinical data have shown an inverse association between circulating sclerostin levels and mortality in dialysis patients [7], concerns have been raised about the cardiovascular safety of drugs targeting sclerostin [17]. Furthermore, the ARCH study recently revealed that romosozumab, an antisclerostin antibody, increased the risk of serious adverse cardiovascular events in postmenopausal women with osteoporosis [18].

To investigate our theory that sclerostin, produced by VSMCs in calcified vessels, may spill over to the serum and thereby, next to impacting the bone, prevent further progression of (vascular) calcifications, a rat warfarin model was employed. In contrast to other models of vascular calcification (animal models for CKD, diabetes, or osteoporosis), this model induces vascular calcifications in the absence of underlying bone pathologies. Hence, the disturbed mineral metabolism in CKD, diabetes, and osteoporosis patients (and animal models) seriously complicates the mechanistic understanding of bone–vascular interactions. In our experimental subjects, warfarin exposure prevented the production of biologically active matrix γ-carboxyglutamic acid protein (MGP) [19,20]. MGP is a local inhibitor of vascular calcification and in order to become biologically active, the protein undergoes post-translational γ-glutamyl carboxylation. This process, which takes place in concert with vitamin K (vitK) as a cofactor [21], is prevented by warfarin since it inhibits vitK recycling. Furthermore, Beazley et al. demonstrated that activation of Wnt/β-catenin signaling was involved in warfarin-induced VSMC transdifferentiation, which resulted in calcification of the medial layers of the arterial wall [22]. Nevertheless, the exact role of sclerostin in the VSMC transdifferentiation process, as well as its role in the bone–vascular axis, has not been thoroughly investigated so far in this model. Therefore, in this study, a time-dependent characterization of this model at the level of the vessels and the bone was coupled to an in-depth investigation of sclerostin biology.

## 2. Results

### 2.1. Mortality, Body Weight, and Serum Markers of Mineral Metabolism and Renal Function in Control Versus Warfarin-Exposed Rats Sacrificed after 10 Weeks

Induction of vascular calcification in rats by dietary exposure to warfarin resulted in low overall mortality of 6.7%. Two warfarin-exposed rats died, out of which one animal died 40 days and the other animal 2 days before the planned sacrifice at week 10 (the latter animal was included in the analyses). Body weight; food intake; and serum calcium, phosphorous, and creatinine were compared between control and warfarin-exposed rats at week 10 and did not differ between control and warfarin-exposed rats (Table 1). Serum creatinine levels were significantly increased in warfarin versus control rats, however, this increase was probably too mild to be considered a compromised renal function, which was further substantiated by the normal calcium and phosphorous levels.

### 2.2. Osteochondrogenic Versus Adipocytic Transdifferentiation of VSMCs

To study VSMC transdifferentiation, relative quantitative proteomic analysis of aortas extracted from control rats and rats exposed to warfarin for 10 weeks was performed. Using stringent and significant cut-off criteria for protein identification (99th percentile peptide identification confidence), we found that warfarin exposure caused significant (*p* < 0.05) differential regulation of 275 proteins (123 upregulated, 152 downregulated) in the aorta. Using latent semantic analysis (LSA)-based interrogation of this list of 275 proteins, employing either Wnt/Catenin or PPAR as the investigating concept, we identified a strong representation in this dataset of proteins linked to both osteogenic (Wnt/Catenin) and adipogenic (PPAR) transdifferentiation. Figure 1 identifies the proteins extracted from the original list that possessed cosine similarity scores of >0.1 (demonstrating at least implicit text association) using either Wnt/Catenin (Figure 1A) or PPAR (Figure 1B) as the input interrogator. Our data indicated that many of the Wnt/Catenin-associated proteins were downregulated, while the opposite was evident for the PPAR-associated proteins (Figure 1C).

### 2.3. Time-Dependent Development of Vascular Calcification

Determination of the calcium content in the vessels showed a time-dependent increase in vascular calcification in warfarin-exposed rats. Compared with control rats, significant differences in calcium content were seen in the aorta, femoral, and carotid arteries from 6 weeks onwards (Figure 2). 

Thoracic aorta slices stained by von Kossa and their corresponding percentage of calcified area are presented in Figure 3. While there was no evidence of vascular calcification in the aorta of control rats, a gradual time-dependent increase in the percentage von Kossa positive area was noted in the warfarin-exposed groups, which differed significantly from controls from week 8 onwards. Calcifications were located in the media of the aortic wall, mainly surrounding elastic fibers.

### 2.4. Serum Sclerostin Levels

Sclerostin levels gradually increased in a time-dependent manner in the serum of rats exposed to warfarin for 4, 6, 8, and 10 weeks (Figure 4). At week 10, serum sclerostin levels were significantly increased (*p* = 0.0381) compared with control animals.

### 2.5. Vascular Expression of Sclerostin mRNA

Aortic sclerostin (*Sost*) mRNA levels were significantly upregulated (×1.66, *p* = 0.0159) in warfarin-exposed rats compared with control rats at week 10 (Figure 5). 

### 2.6. Histological Association between Vascular Calcification and Sclerostin/Low-Density Lipoprotein Receptor-Related Protein 4 (LRP4) Expression

Figure 6 depicts representative images of von Kossa and immunohistochemical sclerostin/LRP4 stained aortic sections obtained from calcified (10 weeks) warfarin-exposed rats (Appendix A shows the corresponding images of control rats). Furthermore, in order to investigate the clinical relevance of sclerostin-induced effects at the level of the vessels, tissue sections from calcified human aortas were investigated. Both calcified rat and human vessels clearly showed the protein expression of sclerostin. Furthermore, there was strong staining for LRP4 in the calcified vessels. LRP4 is known to anchor sclerostin to the site where it exerts its effects. In noncalcified control sections, LRP4 staining was virtually absent, suggesting that the sclerostin present in noncalcified vessels cannot execute its inhibitory function on the Wnt/β-catenin pathway.

### 2.7. Disturbances in Bone Turnover

The analysis of static and dynamic bone parameters in control and warfarin-exposed (10 weeks) rats is presented in Figure 7. Bone area and mineralized bone area were significantly reduced (both *p* = 0.0095) in rats exposed to warfarin compared with control rats. Measures of osteoblast activity (bone formation), including osteoblast perimeter and osteoid area, were lower (in a nonsignificant manner) in warfarin-exposed rats compared with controls. Bone resorption (osteoclast perimeter) and bone mineralization (osteoid width, bone formation rate, and mineral apposition rate) did not differ between control and warfarin-exposed rats.

### 2.8. Sclerostin Expression in the Bone

To investigate osteocytic sclerostin production, immunostaining for sclerostin was performed in the bone. Microscopic analysis of the staining allowed the clear distinction of sclerostin-positive from -negative osteocytes in the bone. The frequency of sclerostin-positive versus -negative osteocytes was recorded by manually counting in a label-blinded manner the osteocytes in the tibia sections. No differences in the percentage of sclerostin-positive osteocytes were observed between control and warfarin-exposed rats (Figure 8).

## 3. Discussion

Many aspects of the vascular media calcification process remain cryptic. Although being generally accepted as the key event during the development of vascular calcification, knowledge of VSMC transdifferentiation is rather fragmentary. Moreover, at this stage, it is not clear which molecules exactly are key in the linkage between bone and vascular pathology (bone–vascular axis). Since the disturbed mineral and bone metabolism in CKD patients further complicates the understanding of the mechanisms underlying the bone–vascular coupling, an experimental animal model with warfarin-induced vascular calcification was applied in this study. This model gave us the opportunity to investigate vascular calcification and the bone–vascular axis, regardless of CKD-induced disturbed mineral balance and the resulting bone pathology. Indeed, serum calcium and phosphorous levels were similar between control and warfarin-exposed rats. 

During the vascular calcification process, one of the key events is the transdifferentiation of VSMCs [2]. Given the fact that VSMCs are of mesenchymal origin and that transdifferentiation of VSMCs induces soft tissue mineralization, a potential transdifferentiation towards an osteoblastic phenotype seems logical. Nevertheless, by further investigating the transdifferentiation process by means of an unbiased quantitative proteomic approach (isobaric tag for relative and absolute quantitation (iTRAQ) protein labeling and mass spectrometry), we showed that next to osteoblastic transdifferentiation involving Wnt/β-catenin signaling, adipocytic transdifferentiation as evidenced by PPARγ signaling was also evident. The latter finding thus is highly indicative of PPARγ signaling functioning as a regulatory factor during the vascular calcification process. Furthermore, it argues for the local vascular action of sclerostin by regulating the balance between both differentiation pathways, that is, inhibiting Wnt/β-catenin signaling on the one hand and inducing PPARγ signaling on the other. 

An inverse relationship exists in the adipogenic and osteogenic lineage commitment, which may indicate that promoting the adipocyte phenotype may exert antagonistic effects on calcification, which is in line with the fact that PPARγ signaling counteracts vascular calcifications [3,4,23] and our hypothesis that sclerostin may inhibit vascular calcification. 

In our current study, we found that sclerostin was expressed at both the mRNA and protein level in calcified vessels, which is in line with earlier studies [9,11,24]. Furthermore, LRP4 protein expression was strongly upregulated in the calcified vessels. LRP4 functions as an anchor for sclerostin in the bone [25,26]. Recently, we demonstrated that mice lacking functional LRP4 expression profoundly lacked sclerostin-positive osteocytes in contrast to wild-type mice, where >30% of osteocytes stained positive for sclerostin [25]. More importantly, LRP4 transgenic mice showed a similar phenotype compared to that of SOST (the gene encoding sclerostin) knockout mice [25]. The fact that LRP4 is clearly present in calcified vessels while virtually absent in normal vessels indicates that the sclerostin produced in the vessels is likely retained there to act locally. Moreover, the fact that we not only localized LRP4 and sclerostin in rat but also human calcified vessels underlines the clinical relevance of our findings. In addition to being retained in the vessels through LRP4 binding, part of the sclerostin produced in the vessels is likely to spill over to the serum. This was hypothesized earlier in a previous study by our group in which sclerostin was quantified in the serum of hemodialysis patients either clear of or presenting with vascular calcification [7]. This may also explain the gradual increase in serum sclerostin levels of animals with progressing ectopic vascular calcification, as observed in our present study. 

Osteocytic sclerostin production was investigated to ensure that the elevated serum sclerostin levels were due to sclerostin production by calcified vessels and did not originate from the osteocytes, which are the main source of sclerostin. Our finding that no differences were observed in the number of sclerostin-expressing osteocytes between control and warfarin-exposed rats strongly indicates that the bone is not likely to be responsible for the observed increased serum levels of sclerostin. Furthermore, we recently found sclerostin protein in the cell culture medium of in vitro calcifying VSMCs, proving that VSMCs should be able to secrete sclerostin towards the circulation (unpublished results).

Bone histomorphometric analysis of warfarin-exposed animals revealed a mildly, however significantly decreased (mineralized) bone area after 10 weeks of warfarin exposure compared with control rats. This finding is in line with the decreased (although not significant) number of osteoblasts (osteoblast perimeter) and reduced (not significant) amount of associated osteoid material found. Mineralization of the deposited osteoid, however, remained intact, as reflected by the absence of any difference in osteoid width and mineral apposition rate. As indicated by the osteoclast perimeter, bone resorption also did not differ between groups. Given the particular role that has been ascribed to sclerostin at the level of the bone [27], the rather mild effects on the bone observed in our present study can be potentially explained by the significant increase in vascular-derived serum sclerostin levels and are compatible with the contribution of vascular sclerostin to the high incidence of low bone turnover in CKD patients.

This study has a few limitations. Since this study was originally designed to only assess the degree of vascular calcification due to warfarin treatment, no urine was collected. Being able to measure the creatinine content in the urine would have made it possible to calculate the glomerular filtration rate (GFR), which is based on the measurement of urine and serum creatinine concentrations and the urinary volume, this is a more accurate measure of kidney function compared with serum creatinine concentration. Future experiments are needed to investigate to what extent increased renal production of sclerostin, in addition to calcified VSMC, contributes to increased serum levels of this protein. 

Overall, this study demonstrated that the sclerostin biology in an animal model of vascular calcification is in agreement with the role of this protein in the vascular calcification process and the pathologically disturbed bone–vascular axis in patients suffering from vascular calcifications. Similar to its function in bone, it could be assumed that sclerostin is produced as a negative feedback mechanism to prevent excessive calcification. These results urge the investigation of the vascular safety of antisclerostin antibody treatment, which currently is ongoing in our laboratory. Hence, antisclerostin treatment is being developed as a bone anabolic treatment for patients with osteoporosis, who are at the same time at risk for vascular calcifications. 

## 4. Materials and Methods 

### 4.1. Statement of Ethics

All experimental procedures were conducted in compliance with the National Institutes of Health Guide for the Care and Use of Laboratory Animals 85-23 (1996) and approved by the University of Antwerp Ethical Committee for Animal Experiments. The experimental procedure involving human arterial sections was approved by the ethics committee of the University of Antwerp (A03 043) on 18 December 2014.

### 4.2. Animals

Thirty 8-week-old male Wistar rats (Iffa Credo, Brussels, Belgium) were housed in standard cages (two per cage) at constant temperature and humidity and exposed to a 12 h light/dark cycle. All rats had free access to tap water and their assigned diet. 

### 4.3. Induction of Vascular Calcification in the Rat Model

The animals were randomly assigned to the following experimental groups: control group (10 weeks (wk), n = 4) and warfarin-exposed groups (4 wk, n = 4; 6 wk, n = 6; 8 wk, n = 8; 10 wk, n = 8). Control rats were fed standard pellet chow for 10 weeks. Rats receiving the warfarin-containing diet (3 mg warfarin/g diet and 1.5 mg vitamin K1/g diet [20], SSNIFF Spezialdiäten, Soest, Germany) were sacrificed at different time points (4, 6, 8, or 10 weeks) in order to evaluate the onset and further progression of vascular calcification.

### 4.4. Human Tissue

Human arterial tissue sections, obtained in the frame of organ donation, were used for histological purposes. After von Kossa staining, four male subjects were identified with severe vascular calcifications. These subjects were then selected for further immunohistochemical evaluation.

### 4.5. Serum Markers of Bone Metabolism and Renal Function

At sacrifice, rats were exsanguinated via the retro-orbital plexus after anesthesia with sodium pentobarbital (60 mg/kg, i.p., Nembutal, Ceva Santé Animale, Libourne, France) via intraperitoneal injection. Serum creatinine was measured according to the Jaffé method. Calcium levels were determined using flame atomic absorption spectrometry (Perkin-Elmer, Waltham, MA, USA) after diluting the sample in 0.1% La(NO_3_)_3_ to prevent chemical interference. Serum phosphorus levels were measured using the Ecoline®S Phosphate kit (Diasys, Holzheim, Germany). 

### 4.6. Evaluation of Molecular Signaling Pathways Involved in VSMC Transdifferentiation

Mass spectrometry and quantitative proteomics (using iTRAQ labeling) were performed on the aortic samples of control rats and rats exposed to warfarin for 10 weeks. The aorta samples were ground completely in a protein extraction buffer (8 M urea, 2 M thiourea, 0.1% SDS in 50 mM triethylammonium bicarbonate solution). The concentrations of the proteins extracted from the aorta were quantified using the reducing agent and detergent compatible (RCDC) protein assay kit (Bio-Rad, Hercules, CA, USA). Equal amounts of proteins from each sample were reduced and alkylated by tris-2-carboxyethyl phosphine and 5-methyl-methanoethiosulphate, respectively, before trypsin digestion. The resulting peptides from each sample were labeled using iTRAQ reagents (Sciex, Redwood City, CA, USA) following the manufacturer’s instructions. To improve LC-MS/MS proteome coverage, samples were subjected to a 2D-LC fractionation system (Dionex ULTIMATE 3000, ThermoScientific, Waltham, MA, USA). The mixed peptides were first fractionated on a strong cationic exchange chromatography polysulfoethyl aspartamide column (1 × 150 mm, (Dionex)) and secondly separated on a nano-LC C18 column (200 Å, 2 μm, 75 μm × 25 cm (Dionex)). The nano-LC was coupled online to a QExactive™-Plus Orbitrap (ThermoScientific) mass spectrometer. The nano-LC eluents were infused to the Orbitrap mass spectrometer with a capillary at 1.7 KV on a nanoelectrospray ionization (nano-ESI) source at a flow rate of 300 nL/min. Data-dependent acquisition in positive ion mode was performed for a selected mass range of 350–1800 *m*/*z* at MS1 level (140,000 resolution) and MS2 level (17,500 resolution). The raw data were analyzed by Proteome Discoverer 2.0 software (ThermoScientific) using Sequest HT as search engine against the rattus norvegicus UniProt/SwissProt database with a threshold of confidence above 99% (false discovery rate less than 1%). The list of identified proteins containing iTRAQ ratios of expression levels over control samples was then generated.

### 4.7. Bioinformatic Analysis

Proteins identified according to the statistical MS cut-offs described previously were then subsequently used for bioinformatic analyses. To identify the significantly altered proteins (i.e., proteins differentially expressed due to warfarin exposure), raw iTRAQ ratios (control:warfarin) were first log_2_ transformed. Following log_2_ ratio transformation, differentially expressed protein (DEP) lists were created by identifying only proteins that possessed log_2_-transformed iTRAQ ratios two standard deviations (*p* < 0.05) from the calculated mean background expression variation level. Significant DEP lists (comprising proteins elevated or reduced in their expression in response to warfarin) were then employed for further bioinformatics deconvolution using diverse informatics platforms including Ingenuity Pathway. LSA of the warfarin DEP lists was performed as previously described using GeneIndexer [28]. LSA is a computational natural language processing investigation technique capable of elucidating latent prosaic connections between interrogator concepts (e.g., biomedical phrases such as PPAR) and biomedical terms (e.g., a specific protein gene symbol).

### 4.8. Evaluation of Vascular Calcification

At sacrifice, aorta, left femoral, and left carotid arteries were isolated. For histological analyses, thoracic aortae were fixed in neutral buffered formalin for 90 min and cut into 2–3 mm thick rings. These rings were then embedded upright in paraffin, of which 4 μm sections were cut and stained for calcification with von Kossa and counterstained with hematoxylin and eosin (H&E). Calcification was evaluated using Axiovision image analysis software (Release 4.5, Carl Zeiss, Oberkochen, Germany). Two color thresholds allowed separation of the calcified area (von Kossa positive, black areas) and the remaining tissue area. The percentage calcified area is defined as the von Kossa positive area versus the total tissue area (von Kossa positive area plus the remaining tissue area).

To determine the calcium content, the proximal part of the abdominal aorta, the left carotid, and femoral arteries were weighed on a precision balance. The tissue samples were then digested overnight in 65% HNO_3_ at 60 °C, followed by dilution in 0.1% La(NO_3_)_3_ to eliminate chemical interference during flame atomic absorption spectrometry. Results were expressed as mg calcium/g wet tissue.

### 4.9. Evaluation of Bone Metabolism

To allow histomorphometric analysis of dynamic bone parameters, all animals received an intraperitoneal injection of 30 mg/kg tetracycline and 25 mg/kg demeclocycline 7 and 3 days before sacrifice, respectively. At sacrifice, the left tibia was isolated and fixed overnight in 70% ethanol. After dehydration and embedding in 100% methyl methacrylate (Merck, Hohenbrunn, Germany), 5 µm thick sections were stained by the method of Goldner for quantitative histology to determine static bone parameters using Axiovision image analysis software (Release 4.5, Carl Zeiss, Oberkochen, Germany). Total bone area, mineralized bone area, osteoid width and area, osteoblast perimeter, eroded perimeter, osteoclast perimeter, and trabecular number were measured using this software platform. Unstained sections of the tibia (10 µm thick) were mounted in 100% glycerol for fluorescence microscopy and visualization of tetracycline and demeclocycline labels. Based on the length and distance between double tetracycline/demeclocycline labels, dynamic bone parameters, including bone formation rate and mineral apposition rate, were measured. 

### 4.10. Identification of Vascular Sclerostin mRNA

Total mRNA was extracted from the distal part of the abdominal aorta using the RNeasy Fibrous Tissue Mini Kit (Qiagen, Hilden, Germany) and complementary DNA was generated with the High-Capacity cDNA Archive Kit (Applied Biosystems, Foster City, CA, USA). Real-time PCR amplification was performed based on the TaqMan fluorescence methodology (ABI Prism® 7000 Sequence Detection System, Applied Biosystems). TaqMan probes and primers for glyceraldehyde 3-phosphate dehydrogenase (GAPDH) (Rn99999916_s1) and sclerostin (SOST) (Rn00577971_m1) were purchased from Applied Biosystems as a Taqman® gene expression assay on demand. For each sample, the expression of the tested transcripts was analyzed in triplicate and normalized to the expression of GAPDH, the housekeeping gene. The comparative CT method, using aortas of control rats as calibrator samples, was used to calculate the gene expression levels.

### 4.11. Immunohistochemical Sclerostin/LRP4 Staining on Rat and Human Arterial Sections

Immunohistochemical staining was performed on neutral buffered formalin-fixed, deparaffinized arterial sections of warfarin-exposed and control rats sacrificed at week 10, as well as on calcified human arterial sections. The excised tissue sections were blocked with normal goat serum (20% in PBS) for 20 min and incubated overnight with polyclonal rabbit antisclerostin (1:500, ab-63097, Abcam, Cambridge, UK) or LRP4 (1:75, sc-98775, Santa Cruz, TX, USA). Biotinylated goat anti-rabbit (Vector Laboratories, Burlingame, CA, USA) was used as a secondary antibody. Avidin/biotinylated peroxidase complex (VECTASTAIN ABC kit, Vector Laboratories) was added as a signal amplifier and 3-amino-9-ethylcarbazole (AEC, Sigma-Aldrich, St. Louis, MO, USA) was used as a substrate. The sections were counterstained with hematoxylin. Sections in which the primary antibody was omitted were used as negative controls. 

### 4.12. Immunohistochemical Staining on Rat Bone Sections

Sclerostin expression was also investigated on tibia sections in order to compare the absolute sclerostin expression per bone area as the percentage of sclerostin-positive osteocytes (lacunae) between control and warfarin-exposed rats. After deacrylation and decalcification of the tibia sections, the same protocol for immunohistochemistry on the arterial sections was used as described above. The antisclerostin primary antibody (ab-63097, Abcam) was used in a 1:750 concentration. 

### 4.13. Evaluation of Serum Sclerostin Levels

The sclerostin concentration in the serum was determined using an antisclerostin ELISA kit (Quantikine® Elisa, R&D Systems, Minneapolis, MN, USA), following the appropriate procedures.

### 4.14. Statistical Analysis

All statistical analyses were performed using PRISM software (GraphPad Prism 6.0, San Diego, CA, USA). Data are presented as the mean ± standard error of mean (SEM). Statistical differences between groups were investigated by a two-tailed Mann–Whitney *U* test. Bonferroni correction was applied when appropriate. *p*-Values < 0.05 were considered statistically significant. 

## Figures and Tables

**Figure 1 toxins-11-00428-f001:**
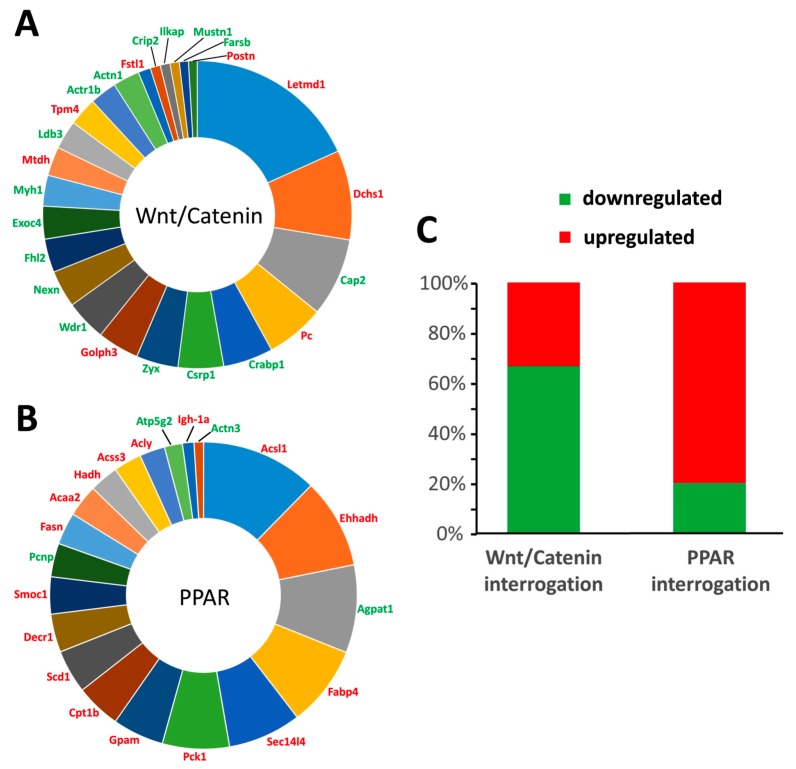
Unbiased informatic interrogation of warfarin-induced aorta proteomic data. Using the input concept interrogators Wnt/Catenin or peroxisome proliferator-activated receptor (PPAR), we identified the proteins from the list of 275 differentially expressed proteins (DEPs) obtained from our isobaric tag for relative and absolute quantitation (iTRAQ) proteomics that possessed a cosine similarity latent semantic analysis (LSA) score of >0.1. This cut-off value for concept-to-text (all available published biomedical abstracts from PubMed Central at NCBI) association indicates at least an implicit association between the input concept (e.g., Wnt/Catenin or PPAR) and the specific term (i.e., gene symbol) found in any of the available biomedical texts. The greater the cosine similarity score (as indicated by the size of the pie chart sector in **A** and **B**) for a given protein indicates the strength of its textual association with the specific input interrogator concept groups (Wnt/Catenin or PPAR). Hence Letmd1 (LETM1 domain-containing protein 1: panel **A**) demonstrates the strongest LSA-based correlation to the Wnt/Catenin interrogator terms while Acsl1 (Acyl-CoA synthetase long-chain family member 1: panel **B**) demonstrates the strongest correlation to the PPAR interrogator terms. The specific proteins annotating the pie charts are depicted in either red (upregulated in warfarin-treated vessels compared to control) or green (downregulated in warfarin-treated vessels compared to control). Panel **C** demonstrates that as a percentage of the total proteins implicitly associated with Wnt/Catenin signaling or PPAR the majority of Wnt/Catenin associated proteins are downregulated compared to control animals by warfarin treatment while the majority of PPAR associated proteins are upregulated compared to control animals.

**Figure 2 toxins-11-00428-f002:**
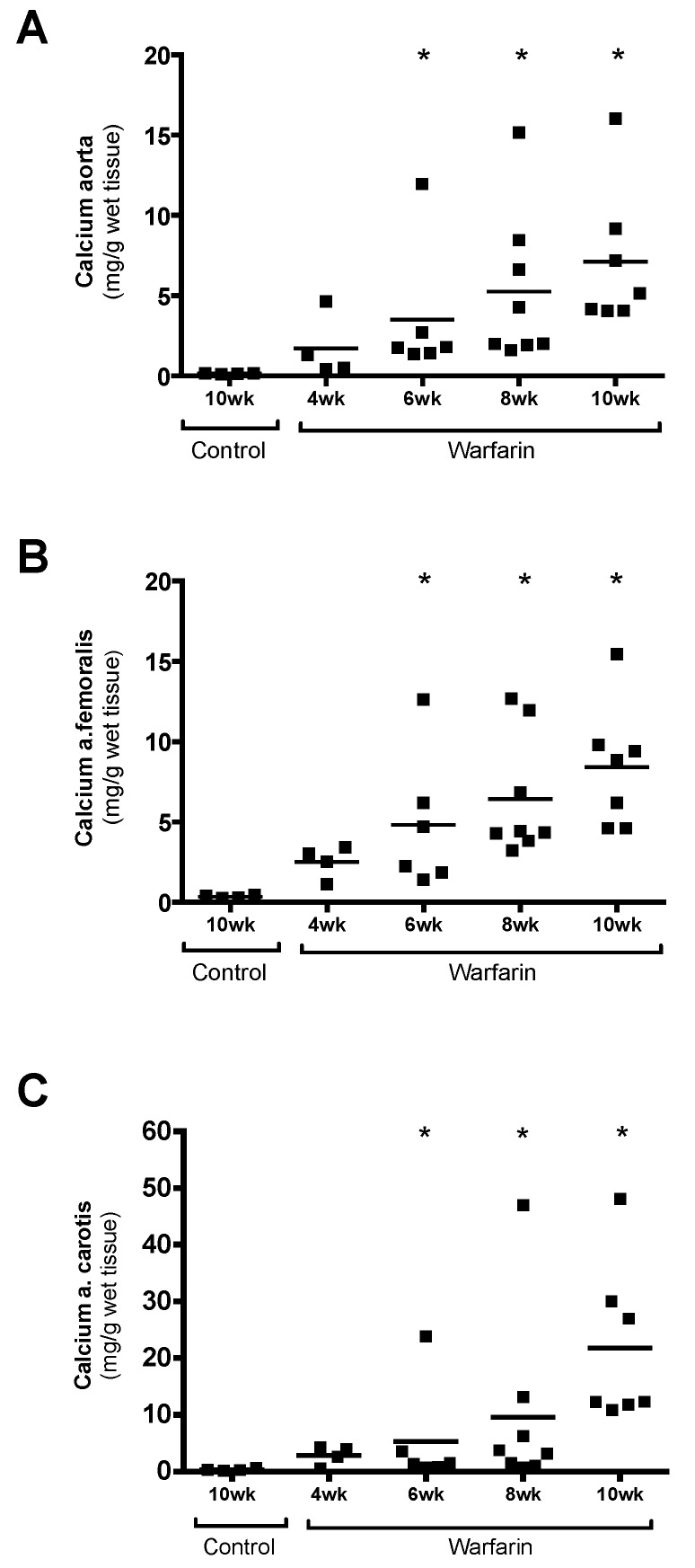
Calcium content. Calcium content of (**A**) the aorta, (**B**) femoral artery, and (**C**) carotid artery of control rats and warfarin-treated rats. * *p* < 0.05 versus control.

**Figure 3 toxins-11-00428-f003:**
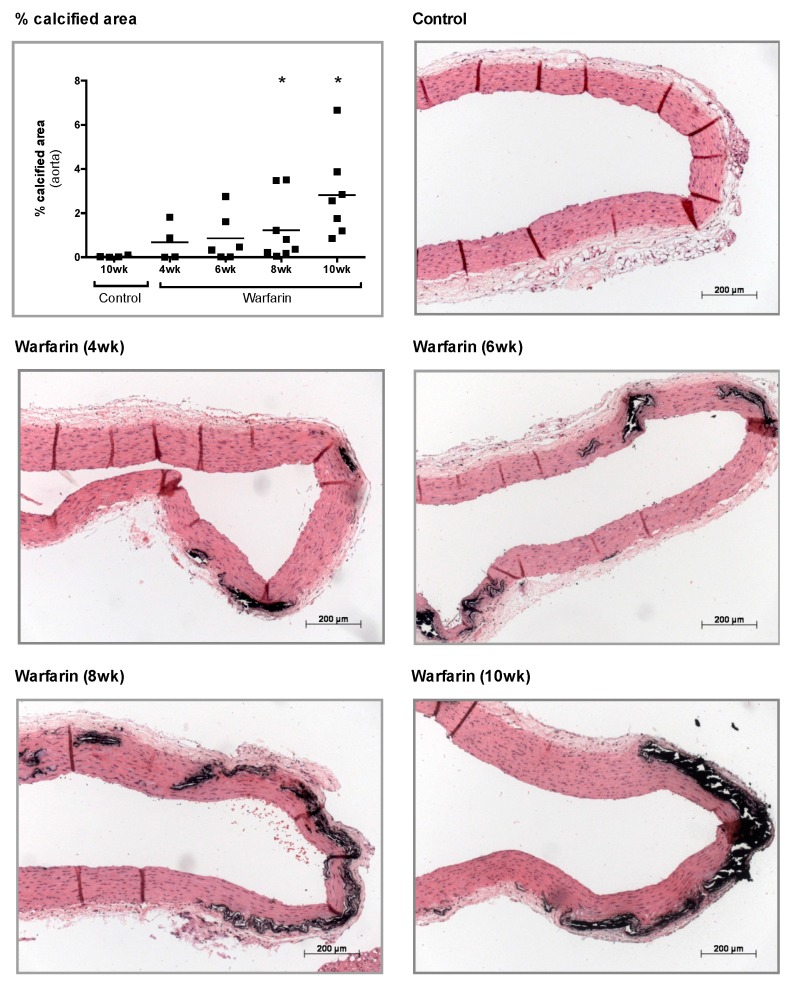
Histological evaluation of vascular calcification. Semiquantitative evaluation of the von Kossa positive area and representative images of von Kossa stained sections of the thoracic aorta, counterstained with hematoxylin and eosin (H&E): control rats and rats treated with warfarin for 4, 6, 8, or 10 weeks. * *p* < 0.05 versus control.

**Figure 4 toxins-11-00428-f004:**
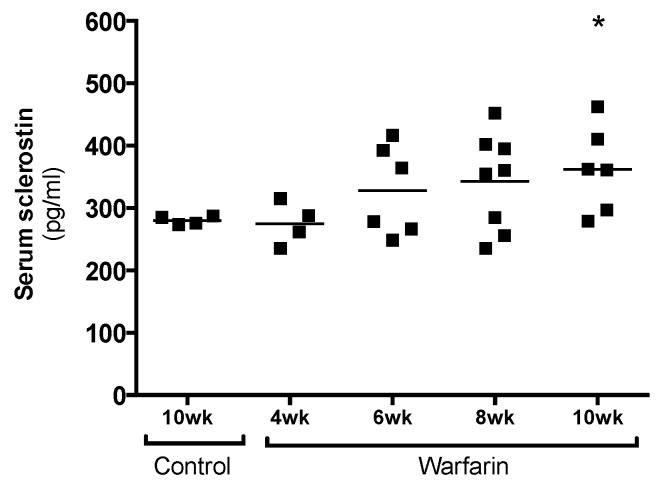
Serum sclerostin levels. Time-dependent increase in serum sclerostin levels after 4, 6, 8, and 10 weeks of warfarin treatment. * *p* < 0.05 versus control.

**Figure 5 toxins-11-00428-f005:**
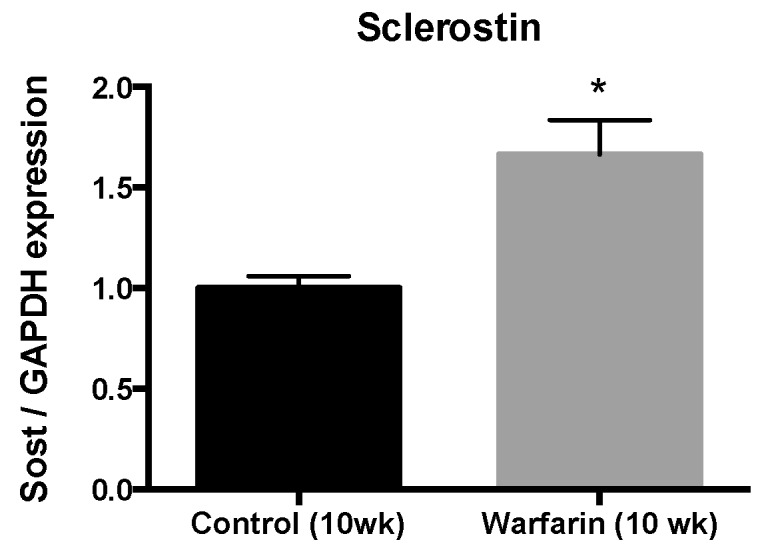
Aortic sclerostin mRNA expression. mRNA expression profile of sclerostin in the abdominal aorta of control rats (10 weeks) and rats treated with warfarin for 10 weeks. * *p* < 0.05 versus control.

**Figure 6 toxins-11-00428-f006:**
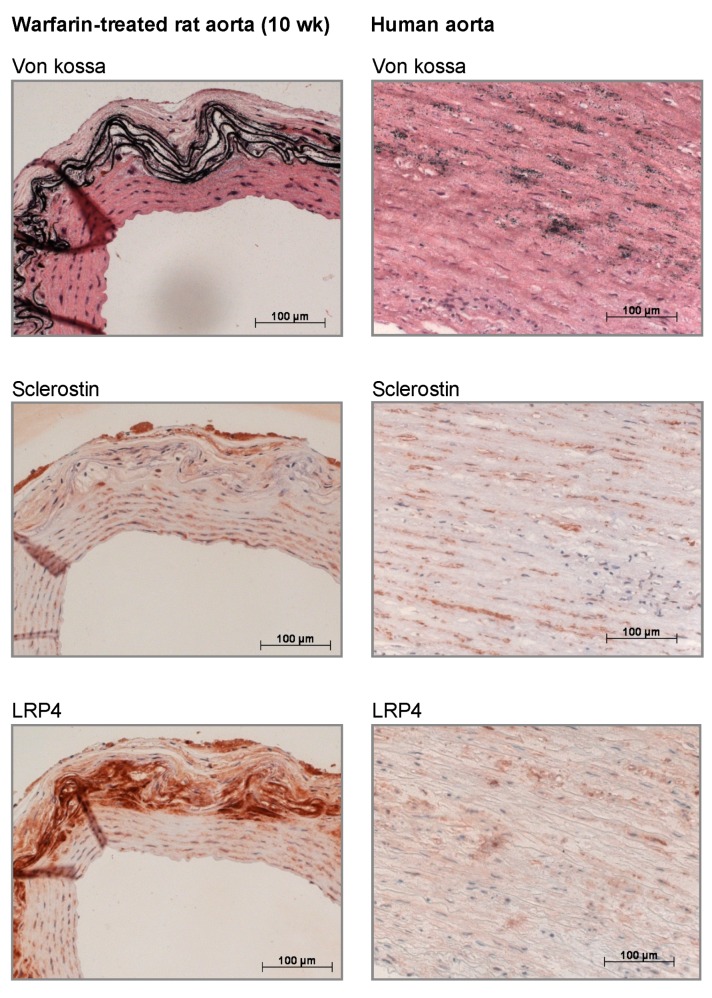
Consecutive tissue sections stained with von Kossa and immunostained for sclerostin and low-density lipoprotein receptor-related protein 4 (LRP4). Tissue sections of a 10-week warfarin-treated rat (**left**) and a calcified human aorta (**right**).

**Figure 7 toxins-11-00428-f007:**
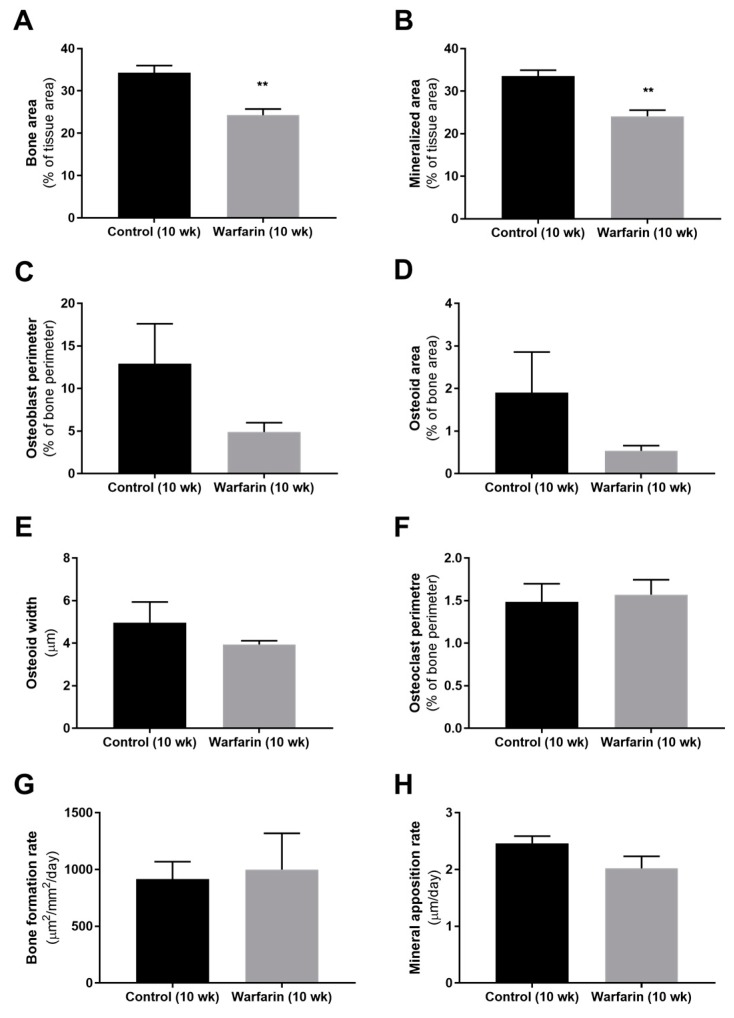
Static and dynamic bone parameters in control rats and rats treated with warfarin. (**A**) Bone area, (**B**) mineralized area, (**C**) osteoblast perimeter, (**D**) osteoid area, (**E**) osteoid width, (**E**) trabecular number, (**F**) osteoclast perimeter, (**G**) bone formation rate, and (**H**) mineral apposition rate. ** *p* < 0.01 versus control.

**Figure 8 toxins-11-00428-f008:**
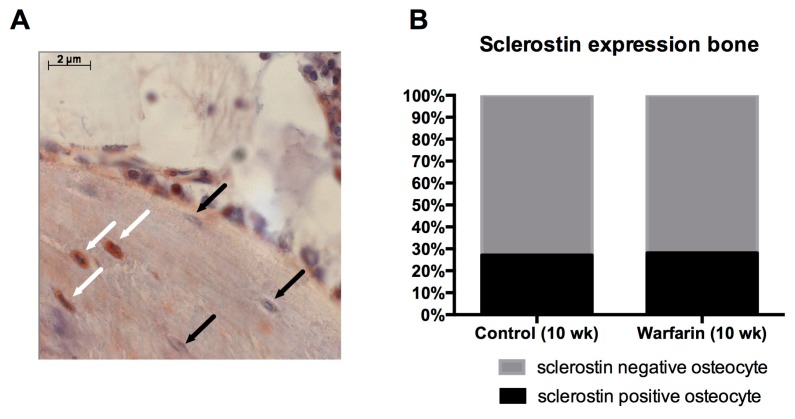
Sclerostin expression in the bone. (**A**) Immunohistochemical staining of sclerostin in the tibia; white arrow: sclerostin-positive osteocyte, black arrow: sclerostin-negative osteocyte. (**B**) Percentage of sclerostin-positive and -negative osteocytes between control rats and rats treated with warfarin.

**Table 1 toxins-11-00428-t001:** Physiological and serum parameters in control and warfarin-treated rats. The data are presented as mean ± standard error of mean (SEM). * *p* < 0.05 vs. control.

Parameter	Control (10 Weeks)	Warfarin Treatment (10 Weeks)
Body Weight (g)	460.5 ± 11.5	440.3 ± 7.9
Food Intake (g/day)	21.5 ± 0.3	20.9 ± 0.3
Creatinine (mg/dL)	0.61 ± 0.02	0.78 ± 0.02 *
Calcium (mg/dL)	11.01 ± 0.19	10.98 ± 0.41
Phosphorous (mg/dL)	3.17 ± 0.16	3.45 ± 0.24

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
