# Peer review of "Sclerostin as Regulatory Molecule in Vascular Media Calcification and the Bone–Vascular Axis"

_toxins, 2019, doi:10.3390/toxins11070428_

Round 1

Reviewer 1 Report

In this study, the authors evidenced the vascular sclerostin production and action on VSMC transdifferentiation and the bone-vascular axis. By using the warfarin-exposed rats model, the authors were able to study vascular calcification in the absence of underlying bone pathologies. They identified the local production of sclerostin in both rat and human calcified vessels. They also confirmed that the increased level of serum sclerostin was originated from excessive local production in calcified vessels in the absence of increased osseous production. These findings provide a novel understanding of the role for sclerostin in the vascular calcification process as well as provide evidence of safety concerns about the anti-sclerostin antibody treatment.

The study was well conducted. Subject definitions and methods were described clearly. Tables and figures were used to present complicated information in a way that is accessible and understandable to the reader. Therefore, I recommend acceptance of this paper for publication.

Author Response

Thank you for the interest in our work. We appreciate your time and effort to carefully read and assess our manuscript.

Reviewer 2 Report

Comments and Suggestions for Authors

The manuscript is an original article conducted on Wistar rats,  but also with implications in human pathology by using human aortas to prove the role of sclerostin in vascular media calcification. Even if sclerostin is a recognized inhibitor of bone formation via Wnt/β-catenin signaling pathway, the article brings new evidence about the paradox of calcification: the deposition of calcium in vessels versus its deprivation at the bone level. The methods used in this research are well documented and enables reproducibility, the immunostained figures are of high quality and also the supplementary figure is suggestive and can be embedded in the paper.

The manuscript is well written in terms of the English language. The topic is of great interest for researchers, but also for clinicians. The manuscript provides a generous introduction and sufficient background to understand its message.

Nevertheless, the study required to improve by reviewing a few minor concerns:

Abstract, line 18: The authors cannot assert with certainty the fact that elevated circulating sclerostin comes from the vessels, so it would be better to add the word      “probably” or an equivalent term (see explanations below at point 3).

Line 48-49:      There is an unfinished phrase.

Line 55:      There are two issues that the authors have to mention.

o   Cejka et al 2014 supported the following hypothesis: elevated serum levels of sclerostin in patients with CKD are not caused by the decreased renal elimination. On the contrary, sclerostin renal elimination increases with declining kidney function because sclerostin was detected also in proximal tubular cells, showing a diffuse cytoplasmic staining pattern.

o   In congruence with previous studies on MGP, as a major vascular calcification inhibitor (line 75-76), it is difficult to envisage for sclerostin exactly which factor interfere with its serum elevation and decrease/increase of renal clearance, and to what extent. Silaghi et al 2019 in Vitamin K dependent proteins in CKD, mentioned some of the factors which were poorly studied: etiology of CKD and the interplay between molecular weight and charge of a molecule in CKD. Sclerostin is a small molecule (approximately 28 KDa) and is positively charged, so it is easy to be filtered by the kidney due to the negative charge of the glomerular membrane.

Consequently,  there are three proven sources for sclerostin: bone, kidney and vessels. Its elevated levels found  in serum may be caused either by the high output of VSMCs (proved by the authors) and proximal tubular cells (which has not been discussed), or by the “mildly” compromised renal function induced in rats in the present study which  may interfere with the interplay between molecular weight and charge of the molecule. As a conclusion, please refer to the two articles and ideas mentioned above.

Line      95: The authors also had to include the word “probably” or an equivalent term when they asserted that “serum creatinine levels…..increase is too mild to be considered a compromised renal function”. To evaluate the renal function only by serum creatinine, calcium and phosphorus are not enough to support this statement. If the eGFR is not used for renal function assessment, serum creatinine cannot indicate a compromised renal clearance until after about 50% of the glomeruli are affected. Besides, there was no association found between sclerostin and phosphorus or calcium in multiple regressions (Cejka et al 2014). Therefore, the authors have to reconsider that statement.

Line 68:      The abbreviation for “vascular smooth muscle cells” is VSMCs,  as it first appears in the manuscript at line 31 and has to be used throughout the entire manuscript consistently.

Line      156: Provide the full term for LRP4 before the abbreviation.

Discussion: The authors do not mention the week points of the study (for example the fact that renal function was not sufficiently investigated - serum creatinine was used instead of eGFR). Also, it would be interesting for future work to assess the extent of each source of sclerostin (bone, kidney, vessels) on its circulating pool.

Line      328 and 343: The unit of measure for the paraffin-embedded rings was probably  4µm and 5µm, respectively, not 4 or 5 m as it appears in the text. Please modify accordingly.

Line 387: the P values < 0.05 were calculated at two tails or at one tail? Please mention.

Please provide an abbreviations list, because it would be easier to follow the text.

Author Response

Thank you for the interest in our work. We appreciate your time and effor to carefully read and assess our manuscript and to give us constructive comments and suggestions, which undoubtedly have improved the scientific quality of our paper.

Abstract, line 18: The authors cannot assert with certainty the fact that elevated circulating sclerostin comes from the vessels, so it would be better to add the word “probably” or an equivalent term (see explanations below at point 3).

We agree with this suggestion and included the term “probably” in the text.

Line 48-49: There is an unfinished phrase.

This is correct, we’ve completed this sentence.

Line 55: There are two issues that the authors have to mention.

Cejka et al 2014 supported the following hypothesis: elevated serum levels of sclerostin in patients with CKD are not caused by the decreased renal elimination. On the contrary, sclerostin renal elimination increases with declining kidney function because sclerostin was detected also in proximal tubular cells, showing a diffuse cytoplasmic staining pattern.

Thank you for this suggestion, we’ve included this in the introduction.

In congruence with previous studies on MGP, as a major vascular calcification inhibitor (line 75-76), it is difficult to envisage for sclerostin exactly which factor interfere with its serum elevation and decrease/increase of renal clearance, and to what extent. Silaghi et al 2019 in Vitamin K dependent proteins in CKD, mentioned some of the factors which were poorly studied: etiology of CKD and the interplay between molecular weight and charge of a molecule in CKD. Sclerostin is a small molecule (approximately 28 KDa) and is positively charged, so it is easy to be filtered by the kidney due to the negative charge of the glomerular membrane.

In line with the reviewer’s pertinent comment, a sentence dealing with this issue was incorporated.

Consequently,  there are three proven sources for sclerostin: bone, kidney and vessels. Its elevated levels found  in serum may be caused either by the high output of VSMCs (proved by the authors) and proximal tubular cells (which has not been discussed), or by the “mildly” compromised renal function induced in rats in the present study which  may interfere with the interplay between molecular weight and charge of the molecule. As a conclusion, please refer to the two articles and ideas mentioned above.

We agree with the reviewer’s comment and included both references in the manuscript.

Line 95: The authors also had to include the word “probably” or an equivalent term when they asserted that “serum creatinine levels…..increase is too mild to be considered a compromised renal function”. To evaluate the renal function only by serum creatinine, calcium and phosphorus are not enough to support this statement. If the eGFR is not used for renal function assessment, serum creatinine cannot indicate a compromised renal clearance until after about 50% of the glomeruli are affected. Besides, there was no association found between sclerostin and phosphorus or calcium in multiple regressions (Cejka et al 2014). Therefore, the authors have to reconsider that statement.

We agree that serum creatinine levels are not the best measure for kidney function. Unfortunately, given the initial setup of this study, no urine collections were performed, which makes that the GFR was not measured. We’ve included this limitation in the discussion section.

Line 68: The abbreviation for “vascular smooth muscle cells” is VSMCs,  as it first appears in the manuscript at line 31 and has to be used throughout the entire manuscript consistently.

OK

Line 156: Provide the full term for LRP4 before the abbreviation.

OK

Discussion: The authors do not mention the week points of the study (for example the fact that renal function was not sufficiently investigated - serum creatinine was used instead of eGFR).  Also, it would be interesting for future work to assess the extent of each source of sclerostin (bone, kidney, vessels) on its circulating pool.

This is a pertinent comment, we’ve included these suggestions in the discussion.

Line 328 and 343: The unit of measure for the paraffin embedded rings was probably  4µm and 5µm, respectively, not 4 or 5 m as it appears in the text. Please modify accordingly.

OK

Line 387: the P values < 0.05 were calculated at two tails or at one tail? Please mention.

OK (two-tailed)

Please provide an abbreviations list, because it would be easier to follow the text.

OK